# Proximate and distant determinants of maternal and neonatal mortality in the postnatal period: A scoping review of data from low- and middle-income countries

Preston Izulla[1], Angela Muriuki[2], Michael Kiragu[1], Melanie Yahner[3‡]*, Virginia Fonner[1,4‡], Syeda Nabin Ara Nitu[5‡], Bernard Osir[1‡], Farahat Bello[3‡], Joseph de Graft-Johnson[3‡]

1 Adroitz Consultants Limited, Nairobi, Kenya, 2 Save the Children, Kenya Regional Office, Nairobi, Kenya, 3 Department of Psychiatry and Behavioral Sciences, The Medical University of South Carolina, Charleston, South Carolina, United States of America, 4 Department of Health and Nutrition, Save the Children, Dhaka, Bangladesh, 5 Department of Global Health, Save the Children Federation Inc, Washington DC, United States of America

☯ These authors contributed equally to this work.
‡ MY, VF, SNAN, BO, FB and JGJ also contributed equally to this work.
* myahner@savechildren.org

**Data Availability Statement:** As this is a scoping review, the authors do not have primary data to make available. All data included are technically

## Abstract

Global maternal and neonatal mortality rates remain unacceptably high. The postnatal period, encompassing the first hour of life until 42 days, is critical for mother-baby dyads, yet postnatal care (PNC) coverage is low. Identifying mother-baby dyads at increased risk for adverse outcomes is critical. Yet few efforts have synthesized research on proximate and distant factors associated with maternal and neonatal mortality during the postnatal period. This scoping review identified proximate and distant factors associated with maternal and neonatal mortality during the postnatal period within low- and middle-income countries (LMICs). A rigorous, systematic search of four electronic databases was undertaken to identify studies published within the last 11 years containing data on risk factors among nationally representative samples. Results were synthesized narratively. Seventy-nine studies were included. Five papers examined maternal mortality, one focused on maternal and neonatal mortality, and the rest focused on neonatal mortality. Regarding proximate factors, maternal age, parity, birth interval, birth order/rank, neonate sex, birth weight, multiple-gestation, previous history of child death, and lack of or inadequate antenatal care visits were associated with increased neonatal mortality risk. Distant factors for neonatal mortality included low levels of parental education, parental employment, rural residence, low household income, solid fuel use, and lack of clean water. This review identified risk factors that could be applied to identify mother-baby dyads with increased mortality risk for targeted PNC. Given risks inherent in pregnancy and childbirth, adverse outcomes can occur among dyads without obvious risk factors; providing timely PNC to all is critical. Efforts to reduce the prevalence of risk factors could improve maternal and newborn outcomes. Few studies exploring maternal mortality risk factors were available; investments in population-based

already publicly available, as the data were extracted from published literature. Original sources are available through the references and this paper represents the full findings of the scoping review.

**Funding:** Financial support for the research, authorship, and publication of this article was received through a grant from the Bill & Melinda Gates Foundation [INV-010128]. The findings and conclusions contained within are those of the authors and do not necessarily reflect the positions or policies of the Bill & Melinda Gates Foundation. The funders had no role in study design, data collection and analysis, decision to publish, or preparation of the manuscript.

**Competing interests:** The authors have declared that no competing interests exist.

studies to identify factors associated with maternal mortality are needed. Harmonizing categorization of factors (e.g., age, education) is a gap for future research.

## Introduction

Maternal and neonatal mortality rates in low- and middle-income countries (LMICs) remain unacceptably high. In 2020, the global maternal mortality ratio was 223 deaths per 100,000 live births, far above the Sustainable Development Goals (SDGs) target of 70/100,000 h1]. Regional disparities persist, with Sub-Saharan Africa accounting for 70% of deaths [1]. Similar trends exist in neonatal mortality. In 2019, the global neonatal mortality rate was 17 deaths per 1,000 live births against the SDG target of 12/1000, with the highest rates seen in Sub-Saharan Africa and Southern Asia–both accounting for 80% of neonatal deaths [2].

The postnatal period, from the first hour of life up to 42 days, is a critical time for both mother and baby, when most deaths occur [3]. Nearly two-thirds of neonatal deaths occur on the first day of life, rising to almost three-quarters by the end of the first week [4]. Nearly half of postnatal maternal deaths occur within the first 24 hours after delivery, and 66% occur during the first week [3]. Timely identification and appropriate management of life-threatening conditions in the postnatal period play a key role in averting preventable mortality [3]. However, while coverage of postnatal care (PNC) varies significantly across countries, it is generally low. Victora et al.'s review reported that a median of 58% of women and 28% of newborns in LMICs received PNC within 42 days after delivery [5]. PNC coverage is lowest during the early postnatal period, when most mothers and babies die, with only 9% of mother-baby dyads in LMICs receiving PNC within 2 days [6–8].

In many LMICs, improving PNC coverage, particularly during the early postnatal period, is challenging due to individual, household, and system-level factors [9]. Key barriers to PNC coverage include high rates of home deliveries in some settings, desire for mothers to leave facilities soon after an uncomplicated delivery due to limitations of staffing and infrastructure [10], and limited coverage of community health worker (CHW) cadres to provide timely home visits [11]. In resource-constrained settings, a potentially effective approach towards reduction of mortality could be identification and follow-up of at-risk mother-baby dyads [12]. Such an approach would require the identification of risk factors (both proximate and distant) that predict the probability of mothers and babies dying in the postnatal period and target at-risk mothers and babies for priority care [13, 14].

We conducted this scoping review to determine the proximate and distant factors that increase the risk of dying in the postnatal period for mothers and their babies. These factors can help identify mother-baby dyads facing an increased risk of mortality during the postnatal period for targeted PNC.

## Methodology

### Search strategy and selection criteria

We applied the Joanna Briggs Institute (JBI) methodology for undertaking scoping reviews [15]. An iterative approach to the selection of included studies was undertaken whereby the inclusion criteria were refined further upon reviewing initial search outputs.

### Search strategy

A team of four individuals with expertise in maternal and newborn health constructed the search strategy. An initial search was undertaken in PubMed, Scopus, Cumulative Index of

Nursing and Allied Health (CINAHL), and PsycINFO databases. Specific Medical Subject Headings (MeSH) and keywords were modified for use in each of the databases searched (search terms are provided in S1 File). The search was limited to studies from LMIC settings published during an 11-year period from January 2011 to December 2022, in line with global guidance for scoping reviews [15].

## Study selection

Primary title and abstract screening were conducted to ascertain relevance by primary screeners. The selected list of publications was subjected to a full-text review to determine eligibility for inclusion. Discrepancies in categorization were resolved through consensus through meetings among up to four secondary screeners.

Table 1 presents the inclusion criteria, and the rationale.:

The following exclusion criteria were applied:

- Intervention studies

- Non-LMIC settings

- Studies that did not report a test of association between risk factors and outcome of interest

- Studies that were not nationally representative

- Facility-based studies

The quality of the studies was assessed using the Newcastle-Ottawa scale for Observational Cohort and Cross–Sectional studies [17]. We used this tool to determine whether the research,

**Table 1. Summary of the inclusion criteria for the scoping review.**

| Study element | Inclusion criteria | Rationale |
|---|---|---|
| Geographical location | Study conducted in a low- or middle-income country as defined by the World Bank | The burden of maternal and neonatal mortality remains highest in LMICs. |
| Coverage | Nationally representative studies | We looked at nationally representative studies to obtain generalizable risk factors. |
| | | We sampled 10% of excluded subnational studies and determined that there were no 'lost' factors, because both identified similar or the same risk factors. |
| Populations | Postnatal women (up to 6 weeks after delivery) and neonates (up to 28 days after delivery) | These are the groups most at-risk for maternal and neonatal mortality. |
| Study period | January 2011- November 2022 | Given the breadth and depth of studies conducted prior to 2011 and the interventions put in place to address potential risk factors over the years, particularly in the years leading up to the end of the Millennium Development Goals era, we sought to identify changes in risk factors in the last decade. |
| Language | Documents in English or those with available English translations | English was the first language for all members of the study team and thus easier to analyze and report on papers in this language. |
| Study designs /methods | **Study designs:** Observational studies (cohort, case-control and cross-sectional) that have a test of association between the risk factor and maternal and neonatal mortality outcome | We looked at representative sampling with a test of association to rule out any confounders. |
| | **Sampling**: Community sample representative of some specified population or a sample with a representative comparison group (i.e., involving matching on some criteria, not a sample of convenience) [16] | |
| | **Data analysis methods:** Test of association of the risk factor and maternal and neonatal mortality outcome | |
| Study outcomes | The main study outcome was death of mother or newborn during the postnatal period. The postpartum (or postnatal) period is defined as the period between birth and six weeks following birth or delivery. | Previous studies have shown that the majority of maternal and neonatal deaths occurred within this six-week window and interventions around this period are likely to be most impactful. |

as applicable: 1) clearly defined the research question, 2) described the study participant selection process including demographics, location, and time period, 3) recruited participants from the same population with uniform eligibility criteria, 4) presented the reasons for the sample size used, 5) assessed exposure of risk factor prior to measuring the outcome, 6) allowed for sufficient time for the observation of outcomes, 7) measured the different levels of exposure with detailed definition, 8) measured exposure to risk factor more than once, 9) defined the outcomes in detail, 10) blinded the outcome measures, 11) had higher follow-up rates, and 12) measured and adjusted for confounding variables. An Excel sheet was used to develop a tracking template of the presence of each criteria, with a total possible score of 14. All studies scored a score of 9 and above. No studies were excluded from the review on the basis of this assessment.

## Data extraction

Data were extracted from the full text of all eligible publications using a standardized Excel data extraction form that captured author, year of publication, objectives, study design, analysis, findings, and additional observations around relevance. An evidence map in Excel that included the studies with the statistical association of the various risk factors at univariate, bivariate, and multivariate analysis was created [18].

## Conceptual framework

To conceptualize the interplay of causative factors and maternal and neonatal mortality, an adapted version of the Mosley and Chen framework was used (Fig 1) [19].

In the framework, distant factors are the broader socioeconomic factors at the individual, household, or community level such as education, wealth status, and residence. These act

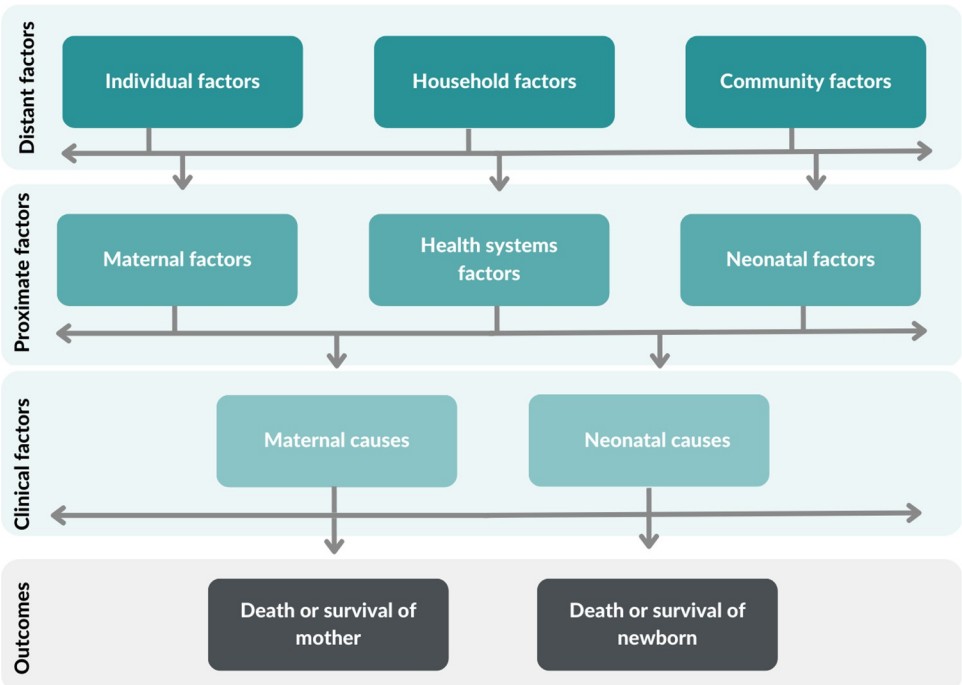

**Fig 1. Proximate and distant factors determining maternal and neonatal mortality, adapted from Mosley and Chen.**

through the proximate factors, which are primarily bio-behavioral factors related to the mother and/or neonate, such as maternal age, birthweight, and utilization of health services that are more directly linked to observed clinical manifestations such as infection or bleeding which led to death. The study team qualitatively identified the common themes emerging from the included papers. A series of virtual meetings were held to categorize risk factors as either distant or proximal. After categorization, data from the extraction tables was summarized into the evidence map.

## Patient and public involvement

Patients and the public were not involved in the design, conduct, reporting, or dissemination plans of the research.

## Results

### Overall findings

Seventy-nine studies were included in the review [20–97]. Of the included studies, 34 studies were conducted in South Asia [21–23, 27, 30–37, 39–41, 46–49, 51, 55, 60, 63, 71, 73–80]; 33 studies were conducted in Sub-Saharan Africa [23, 25, 26, 28–30, 32, 33, 39, 45, 46, 48, 51, 52, 58, 65, 67, 69–71, 75–77, 81, 82, 84, 85, 87, 89, 90, 97]; two were conducted in South America [55, 66]; one was conducted in Sub-Saharan Africa, South America and South Asia [35]; three in South West Asia [42, 62, 93]; one in South Asia, South West Asia, and Sub-Saharan Africa [91]; two in South America [55, 66]; two were global [60, 83]; one in North Africa [74]; and one in Eastern Europe [34].

Five studies [26, 31, 66, 85, 98] explored risk factors for maternal mortality, one study focused on maternal and neonatal mortality while the rest focused on neonatal mortality.

Sixty-eight studies were cross sectional studies [22–33, 36, 37, 39–56, 58–65, 67–72, 75, 76, 78–83, 85–94, 96–98], four studies were retrospective studies [66, 73, 74, 84], four studies were cohort studies [20, 21, 35, 95], one study was a case control study [57], one study was a desk review [34], and one study was a panel study [77].

Fig 2 below summarizes the selection process for papers included in the study.

We noted gaps in standardized categorization and reference points for the various risk factors that not only made it difficult to compare findings across different studies, but also influenced the direction of the observed relationship. For example, using maternal age as a risk factor, the direction of the relationship depends on the reference and comparison age groups, both of which vary significantly across studies limiting comparability.

In Table 2 below, we summarize the risk factors that reported an association between the respective factor and maternal and neonatal mortality at univariate and multivariate analysis.

### Proximate factors

**Maternal factors.** *Maternal age.* Thirty-nine studies [20, 22, 23, 26, 29, 30, 33, 34, 37, 41–43, 45, 47, 49, 51, 52, 56, 58–62, 64, 65, 67, 72, 75, 77, 79–82, 85, 89, 93, 96] found an association between maternal age and maternal and neonatal mortality. The studies used different age bands for comparison, which makes it challenging to present comparison age bands in the results. Nineteen studies [20, 23, 28, 30, 37, 42, 45, 49, 51, 59, 60, 75, 79–82, 96, 97] found an increased risk of mortality among neonates of mothers aged under 20 years compared to those older than 20. A further ten [20, 24, 37, 47, 59, 60, 79, 85, 93, 97] reported an increased risk of neonatal mortality in those aged between 20 and 34. However, these studies used varying reference age ranges, with five studies [20, 37, 59, 60, 97] using 20–24, five [29, 33, 43, 58, 64]

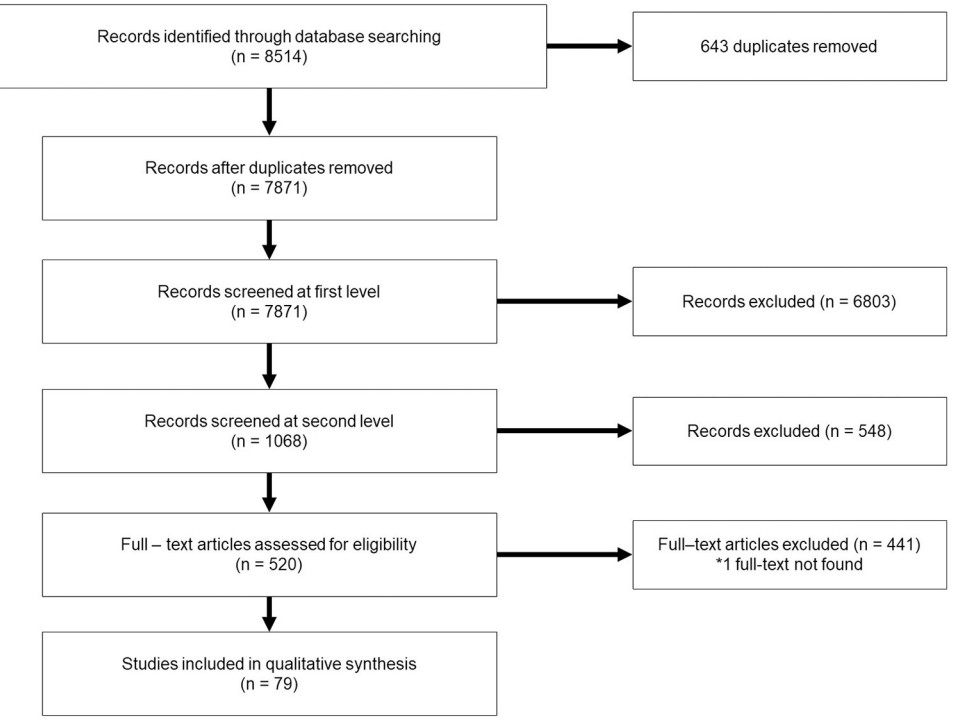

**Fig 2. Flowchart of included studies.**

studies using 19–24 and less than 24 respectively, and four studies [30, 51, 76, 97] using 30–34. Fourteen studies [20, 28, 29, 33, 34, 42, 47, 51, 60–62, 65, 75, 77] reported an increased risk of neonatal mortality for mothers aged above 35 years compared to those below 35. Seven studies found an increased risk of maternal mortality at older age [29, 30, 34, 51, 60, 62, 82].

*Parity, birth rank, birth order, and birth interval.* Nineteen studies [22, 28–30, 33, 34, 37, 43, 44, 51, 53, 56, 59, 61, 64, 75, 77, 82, 87] found an increased risk of mortality for neonates born to primiparous women (first birth order/first birth rank). Seven studies [33, 58, 69, 75, 76, 79, 93] found an increased risk in neonates of multiparous women. The categorization of multiparous women ranged from more than three to more than five, making it difficult to compare across the various studies.

Seventeen studies [20, 30, 42, 45, 47, 48, 51, 58, 61, 62, 66, 67, 72, 76, 79, 86, 87] found an association between birth intervals, while one [49] found no association. While each of the studies used different reference ranges for birth intervals, shorter birth intervals resulted in increased mortality. For example, four studies used 18 months as the cutoff for short birth intervals, while 10 used 24 months, with both groups observing higher mortality for intervals less than the cutoff period. Houweling et al. assessed the effect of longer birth intervals and reported an increased risk in neonatal mortality in birth intervals exceeding 68 months [20].

Eight studies [24, 27, 37, 40, 43, 68, 75, 96] found an association between birth order and birth interval as a combined variable. Five [24, 37, 40, 75, 96] found an increased risk in mortality for first order and/or fourth-order or higher neonates with birth intervals of less than 24 months. Two studies [43, 68] found an increased risk in second and third rank with a birth interval less than 24 months compared to other birth ranks with longer intervals. In contrast, Nisar and Dibley found a decreased risk of mortality in fourth or higher order neonates with a birth interval of less than 24 months [68].

**Table 2. Evidence map of risk factors and their association to maternal and neonatal mortality.**

| Category | Risk factor | No association at uni/bi-variate | Association at uni/bi-variate | Association at multivariate | No association at multivariate |
|---|---|---|---|---|---|
| | | **DISTANT FACTORS** | | | |
| Individual (maternal) | Maternal education (or literacy) level* | [29, 33, 48, 63, 69, 83, 90] | [22, 24, 27, 34, 35, 40, 42, 44, 45, 49, 51, 52, 54, 56, 59, 61, 73, 74, 79–82, 85, 86, 91, 92, 95, 96] | [20, 22–24, 34, 35, 37, 43–45, 47, 49, 53, 54, 58, 59, 62, 64, 65, 72, 77, 79–82, 85, 87, 92, 93, 96] | [25, 27, 40–42, 51, 52, 56, 59, 86, 90] |
| | Community education levels among women | | [42, 86] | [42, 58] | [86] |
| | Maternal occupation | [27, 51, 61, 69] | [24, 40, 54, 86, 91] | [24, 40, 54, 86] | [25, 37, 44] |
| | Marital status | [58, 61, 90] | [45, 70, 81, 85] | [26, 65, 81, 93] | [25, 37, 45, 68, 90] |
| | Smoking/use of alcohol | | [49, 58] | [49] | [58] |
| Family (or household) | Household income/ wealth* | [22, 27, 33, 42, 48, 59, 63, 69, 81, 90] | [40, 41, 44, 45, 49, 51, 56, 67, 68, 73, 74, 79, 80, 82, 83, 85, 86, 91, 92, 96] | [29, 41, 44, 49, 68, 72, 79, 82, 85, 86, 92, 93, 96, 97] | [37, 40, 43, 45, 51, 56, 64, 67, 81, 90] |
| | Paternal education | [27, 63, 69] | [24, 40, 42, 54, 91, 92] | [40, 42, 48, 54, 92, 97] | [24, 25, 37, 41] |
| | Paternal working status/occupation | [69] | [40, 41, 68] | [40, 41, 68] | |
| | **Gender dynamics at household level:** | | | | |
| | Intimate partner violence* | | | [39, 93] | |
| | Gender of household head | [58] | [42, 67] | [67] | [42] |
| | Participation in decision making | | [40] | [40] | |
| | **Household environmental conditions:** | | | | |
| | Source of water | [22, 59, 65, 97] | [27, 40, 52, 54] | [47] | [25, 27, 37, 40, 43, 52, 54, 58] |
| | Type/availability of toilet | [65, 97] | [22, 40, 49, 54, 58, 59, 79] | [49] | [22, 25, 37, 40, 43, 47, 54, 59] |
| | Type of cooking fuel* | [37] | [22, 35, 40, 44, 58] | [35, 37, 40, 44, 72, 78] | [22, 43, 68] |
| | Smoking in the house | [35] | | | |
| | Type of house/kitchen | | [54, 58] | | [37, 44, 54] |
| | Media exposure | | [40, 58, 91] | [72, 87] | [40, 58, 67] |
| | Polygamous marriage | [58] | | | [68] |
| Community | Residence rural vs urban* | [29, 33, 42, 53, 61, 63, 72, 90] | [22, 27, 40, 45, 49, 56, 58, 59, 70, 74, 79–83, 85, 91, 92, 95, 96] | [44, 47–49, 51, 52, 75, 77, 79, 81, 82, 84, 85, 87, 92, 93, 97] | [22, 27, 40, 43, 45, 56, 59, 64, 68, 90, 96] |
| | Community wealth/ infrastructure | [42] | [54, 58, 67, 69, 86] | [69, 87] | [54, 67, 86] |
| | Geographic/ administrative region | [22, 24, 51, 59, 61, 65] | [27, 34, 40, 42, 45, 49, 53, 54, 56, 63, 68, 86, 96] | [34, 37, 40, 42, 43, 45, 47–49, 52, 54, 64, 68, 86, 96] | [27, 56, 58] |
| | Religion | [37, 48, 51] | [22, 54, 56, 59, 96] | [22, 47, 59, 96] | [37, 43, 54, 56, 64] |
| | Tribe/caste | [27, 69] | [54] | [37, 43, 54, 99] | [47, 64] |
| | | **PROXIMATE FACTORS** | | | |
| Reproductive | Maternal age at childbirth* | [40, 41, 44, 48, 53, 68, 90, 92, 95] | [22, 27, 28, 34, 35, 41, 42, 44, 45, 49, 54, 56, 59, 60, 67, 74, 79, 81–83, 85, 86, 91, 92, 96] | [20, 22, 23, 26, 29, 30, 33, 34, 37, 41–43, 45, 47, 49, 51, 52, 54, 56, 58–62, 64, 65, 67, 72, 75, 77, 79–82, 85, 89, 93, 96] | [25, 27, 35, 44, 68, 86, 90] |
| | Parity/birth rank/birth order* | [23, 48, 49, 63, 65] | [21, 22, 28, 34, 35, 42, 44, 45, 54, 56, 59, 67, 69, 70, 73, 74, 79, 80, 82, 85, 86, 91, 92, 95] | [20, 22, 23, 26, 29, 30, 33, 34, 37, 41–43, 45, 47, 49, 51, 52, 54, 56, 58–62, 64, 65, 67, 75, 77, 79, 80, 82, 87, 89, 92, 93, 96] | [42, 45, 67, 86] |
| | Birth interval* | [63] | [28, 42, 45, 49, 53, 65–67, 69, 74, 79, 80, 86] | [20, 30, 42, 45, 47, 48, 51, 58, 61, 62, 66, 67, 69, 72, 76, 79, 86, 87] | [49] |
| | Birth interval and birth order (combined) | | [24, 27, 40, 68, 96] | [24, 27, 37, 40, 43, 68, 75, 96] | [25] |
| | Desire for pregnancy | [27, 51, 63, 67, 68] | [42, 81] | [81] | [25, 42] |
| | Previous death of child | [95] | [27, 40, 50, 80, 83] | [27, 40, 50, 58] | |
| | No. of children <5 yrs. | | [24, 40] | [24, 40] | |
| | Contraceptive use | [28] | [27, 58, 85] | [27, 37] | |

*(Continued)*

**Table 2.** (Continued)

| Category | Risk factor | No association at uni/bi-variate | Association at uni/bi-variate | Association at multivariate | No association at multivariate |
|---|---|---|---|---|---|
| Neonatal factors | Size of baby | [41, 45, 49, 83] | [28, 34, 42, 63, 67, 69, 79, 81, 82, 84, 91, 92, 96] | [20, 23, 25, 30, 41, 42, 47, 48, 51, 53, 62, 64, 65, 67–69, 75, 79, 81, 82, 84, 92, 96] | [34] |
| | Sex of baby | [22, 49, 53, 59, 68, 69] | [24, 28, 35, 40, 42, 45, 54, 56, 67, 70, 79, 81–83, 86, 91, 92, 95, 96] | [24, 29, 35, 36, 40, 43, 45, 47, 48, 51, 54, 58, 62, 64, 65, 67, 68, 72, 75, 79–82, 87, 89, 92, 96, 97] | [25, 42, 56, 86] |
| Access to and utilization of health services | Use of ANC* | [29, 37, 40, 83] | [23, 28, 35, 42, 49, 54, 56, 59, 61, 63, 67, 69–71, 73, 84, 85, 91, 92] | [35, 42, 43, 52, 59, 65, 67, 69, 81, 84, 85, 87, 92] | [23, 41, 68] |
| | Tetanus toxoid and iron/folic acid use in ANC | [29, 65] | [24, 27, 45, 49, 54, 61, 67, 69] | [24, 27, 43, 45, 49, 54, 61, 69] | [20, 37, 67, 68] |
| | Place of delivery (home vs facility) * | [22, 27, 29, 35, 51, 54, 59, 65, 83] | [40, 42, 49, 56, 63, 69, 70, 73, 79, 81, 82, 85, 86, 91, 92] | [25, 40, 79, 81, 82, 85, 87, 89, 92] | [20, 35, 37, 41, 42, 49, 54, 56, 86] |
| | Type of facility for delivery (private vs public) | [51] | [45] | [25, 45] | |
| | Skilled birth attendant | [22, 27, 35, 45, 51, 59, 65, 68] | [35, 42, 56, 58, 63, 73, 79, 82, 83, 86, 90, 96] | [35, 57, 79, 90] | [25, 42, 86, 96] |
| | Postnatal care | [49, 63, 86] | [57, 71, 81, 92] | [81, 89, 92] | |
| | Access to insurance/ social protection scheme | | [54, 67] | | [54, 67] |
| Other factors: maternal, gestational, and delivery factors | Gestation (single vs multiple) | | [22, 28, 40–42, 67, 69, 70, 86, 91, 92, 96] | [20, 22, 40–42, 48, 58, 67, 69, 86, 87, 92, 96] | [25] |
| | Maternal nutritional status | [22, 97] | [40] | [33, 40, 55, 61, 75] | [44, 64] |
| | Early initiation of breastfeeding | [49] | [28, 69, 79, 83] | [64, 69, 79, 89] | |
| | Mode of delivery (vaginal/caesarean section) | [45] | [63, 68, 82, 88] | [23, 29, 30, 51, 65, 75, 82, 88] | [20] |
| | Pregnancy/delivery complications | | [28, 29, 54, 68] | [29, 54, 68] | [26] |

* Risk factors associated with maternal mortality

Note that

• The number (count) of studies in any of the columns does not indicate the strength of association due to varying methodologies applied in the different studies.

• Clinical risk factors are not reported since the study excluded facility-based studies. Clinical risk factors are often examined in facility—rather than population—based studies.

• The results section discusses the findings at the multivariate level.

• Few studies meeting inclusion criteria explored factors contributing to maternal mortality. Factors found to contribute to maternal mortality are noted with an asterisk (*).

*Desire for pregnancy*. Of the eight studies [25, 27, 42, 51, 63, 67, 68, 81] that examined desire for pregnancy for pregnancy and risk of mortality, only one [81] found an association. Arunda et al. found that there was the highest risk in neonatal mortality among adolescent mothers who had unintended pregnancies in marital union [81].

*Previous history of death of child*. All four studies [27, 40, 50, 58] that examined the association between history of previous death of a child and mortality in subsequent/later pregnancies found an increased risk of neonatal mortality in subsequent pregnancies. Kapoor et al. observed that it independently predicted a more than two-fold increase in neonatal mortality [50].

*Complications during delivery and mode of delivery*. Three studies [29, 54, 68] found an increased risk of neonatal mortality among mothers who reported any complication during pregnancy or delivery. The single study [26] that looked at maternal mortality did not find an association with pregnancy complications. Six studies [29, 30, 51, 65, 75, 88] found an increased risk of neonatal mortality for mothers who delivered by caesarean section compared to vaginal delivery. Only Fawole et al.'s study found a decreased risk in mothers who had an elective or emergency caesarean section compared to normal delivery [23].

*Maternal nutritional status*. Five studies [33, 40, 55, 61, 75] reported an increased risk of neonatal mortality in obese mothers (Body Mass Index >30) and one paper[30] reported a decreased mortality among underweight mothers.

## Neonatal factors

**Sex of baby.** Twenty-eight studies [24, 29, 35, 36, 40, 43, 45, 47, 48, 51, 54, 58, 62, 64, 65, 67, 68, 72, 75, 79–82, 87, 89, 92, 96, 97] found an elevated risk of mortality among male newborns, with six [22, 49, 53, 59, 68, 69] reporting no association between sex and neonatal mortality. Several authors noted that this was due to biological or endogenous factors that increase risk for the male neonate during the first week of life and that this risk is reversed after the first week due to social or exogenous factors that place female neonates at higher risk.

**Size of baby.** Twenty-two studies [20, 23, 25, 30, 41, 42, 47, 48, 51, 53, 62, 64, 65, 67–69, 75, 79, 81, 82, 84, 92, 96] found an association between the size of the baby and mortality, with most using an estimated size at birth based on the mother's observation. Eighteen [23, 25, 30, 41, 47, 51, 53, 62, 64, 67, 69, 75, 79, 81, 82, 84, 92, 96] found higher odds of death for smaller babies and four [42, 48, 65, 68] found higher odds for large babies.

Obeidat et al. found that, compared to normal birth weight singleton births, low birth weight single births had an over five-fold increased risk of neonatal mortality, which rose to approximately 16-fold low birth weight with multiple gestation [62]. Multiple gestation with normal birth weight had the same outcomes as normal birth weight single gestation.

Four studies [42, 48, 65, 68] found higher odds of neonatal mortality for larger babies compared to smaller babies within the same study. For example, Kibria et al. found that larger babies were three times more likely to die, while smaller babies were two times more likely to die compared to average/normal size [42]. Similarly, Nisar and Dibley found a 70% and 60% increased risk in mortality for larger and smaller babies, respectively, compared to normal size babies [68].

**Multiple gestation.** Thirteen studies [20, 40–42, 48, 58, 59, 67, 69, 86, 87, 92, 96] reported an increased risk of neonatal mortality among multiple pregnancies compared to singleton pregnancies, with only one reporting no association [25].

## Health system factors

**Antenatal care (ANC) attendance.** Thirteen studies [35, 42, 43, 52, 59, 65, 67, 69, 81, 84, 85, 87, 92] found a reduced risk of neonatal mortality in mothers who attended ANC, with three [23, 41, 68] reporting no association. Manjavidze et al. found that even one ANC visit led to a 6% reduction in perinatal mortality compared to no visits [34]. Akter et al. found that an increase in number of ANC visits resulted in further reduction in likelihood of neonatal mortality [63].

Singh et al. found that attending ANC in the first trimester and achieving more than four visits led to a 40% reduction in neonatal mortality, while ANC attendance in first trimester and less than four visits led to a 20% reduction [54]. Those attending ANC in the second or third trimester only had a 10% reduction in neonatal mortality.

Nisar and Dibley found no difference in outcomes with ANC from trained or untrained health providers [68].

**Place of delivery and skilled birth attendant.** Five studies [25, 42, 67, 79, 89] reported better neonatal outcomes for facility delivery compared to home delivery. For example, Zwane and Msango found lower odds of neonatal mortality among mothers who delivered in private facilities in Swaziland compared to public or home deliveries [25].

In contrast, five studies [40, 81, 82, 85, 92] reported higher risks of neonatal mortality in facility delivery compared to home delivery, which they attributed to low levels of facility delivery in the study population with late referrals leading to poor outcomes.

Patel et al. observed that, compared to physician-attended deliveries, those attended by a nurse midwife had a 30% lower risk of perinatal mortality, stillbirths, and early neonatal mortality [35]. Perinatal mortality, which includes neonatal deaths occurring in the first week of life, had a 40% lower risk in deliveries with no skilled birth attendant compared to physician-attended deliveries. Baruwa et al found that neonatal mortality is higher among children delivered by non-skilled birth attendants [90].

## Distant factors

**Individual level factors.** *Maternal education.* Thirty studies [20, 22–24, 34, 35, 37, 43–45, 47, 49, 54, 58, 59, 62, 64, 65, 72, 77, 79–82, 85, 87, 92, 93, 96] found an association between maternal education and maternal and neonatal mortality at multi-variate analysis. The higher the level of education attained the lower the risk of maternal and neonatal mortality observed. For example, in a study in Bangladesh, Kamal observed that when compared to no formal education, primary education reduced neonatal mortality by 28%, which increased to 33% for secondary education and was highest at 85% for those with above secondary education [59]. Asamoah et al. found higher odds of maternal mortality due to infections and unsafe abortions among women with lower than tertiary level of education [26].

The effect of maternal education on reduced neonatal and/or maternal mortality was assessed at community level in three studies [42, 58, 86]. One, Kibria et al., found a 60% reduction in neonatal mortality in communities with more than 50% of women with higher than secondary education [42]. There was no association noted in the other two studies.

*Maternal occupation/income.* Four studies [24, 40, 54, 86] found an increased risk of neonatal mortality if mothers were working in the informal sector e.g., in agriculture or as laborers, with the four showing no statistically significant association at multi-variate analysis. In one study, there was up to 50% reduction in neonatal mortality when the mother was unemployed compared to when the mother was working, highlighting the key role of the mother in the first few weeks of life [68].

*Marital status.* Of the thirteen studies [25, 26, 37, 45, 58, 61, 65, 68, 70, 81, 85, 90, 93] that examined marital status as a risk factor, three found a decreased risk of neonatal mortality when mothers were married [26, 65, 93].

Asamoah et al. reported that marital status influenced the cause of maternal mortality with married women having more incidences of hemorrhage and infections compared to unmarried women who had more incidences of abortion-related mortality [26].

## Household factors

**Household income.** Twelve studies [29, 41, 44, 47, 49, 68, 72, 82, 86, 92, 93, 96, 97] found an association between household income and neonatal mortality with, with households with lower incomes having a higher risk of neonatal mortality compared to those with higher income levels. Ten studies [37, 40, 43, 45, 51, 56, 64, 81, 87, 90] reported no association. One

study [85] found an association between household income and maternal mortality. Two studies, Ghimire et al., and Alam et al. observed that middle-income households had higher risk of neonatal mortality when compared with those that had higher and lower income, but the risk was confounded with use of biomass as household fuel and physical violence respectively [37, 72].

**Paternal education and occupation.** Six studies [40, 42, 48, 54, 92, 97] found that higher levels of paternal education (secondary level or higher) were associated with lower risk of neonatal mortality. Kibria et al. noted that incremental levels of paternal education led to lower levels of neonatal mortality with a 10% reduction observed if the father had primary education, which increased to 30% if the father had above secondary education [42]. Three studies [40, 41, 68] reported an association between neonatal mortality and paternal occupation or working status, which was further influenced by other factors such as sector of employment or working status of mother. Zakar et al. reported up to 64% higher risk of neonatal mortality in households where the father engaged in manual work compared to those where the father was engaged in management [41]. Hossain et al. observed that in Bangladesh, households where the father was in business or in formal (non-agriculture) employment had a 21% lower risk of neonatal mortality compared to those where the father's occupation was agriculture [40]. The effect of paternal employment may be influenced by mother's employment status as noted by Nisar et al. who found a 50% higher risk of mortality when both parents were employed compared to where the father was employed, and the mother was at home [68].

**Household environmental conditions (indoor pollution, drinking water, toilets).** Five studies [35, 37, 40, 44, 72] found that households using polluting fuels such as biomass and solid fuels had higher risk of neonatal mortality compared to those that used clean fuels. Patel et al. found no association between daily smoking in the house and perinatal mortality [35]. Two studies found a higher risk of neonatal mortality in households where there was no toilet compared to those with a toilet [47, 49], while seven found no association [22, 25, 37, 43, 47, 54, 56]. Kc et al., found a 21% higher risk of neonatal mortality in households where there was no toilet compared to those with a toilet [49]. Ram et al., found statistically lower risk of neonatal mortality in households with clean sources of water [47].

**Gender dynamics at household level.** Tessema et al. found lower neonatal mortality in female-headed households compared to male-headed households, while Kibria et al. found no association [42, 67]. Pool et al. found a 90% increase in neonatal mortality among neonates whose mothers had experienced physical violence compared to mothers who had not experienced violence [39] Hossain et al. reported that maternal participation in decision-making was associated with an 11% lower risk of neonatal mortality [40].

## Community level factors

**Rural/urban.** Seventeen studies [44, 47–49, 51, 52, 75, 77, 79, 81, 82, 84, 85, 87, 92, 93, 97] found higher maternal and neonatal mortality in rural communities compared to urban communities, while 11 studies [22, 27, 40, 43, 45, 56, 59, 64, 68, 90, 96] found no difference. None of the studies found a higher risk among urban communities. Asamoah et al. reported higher odds of maternal mortality among women residing in rural areas compared to those in urban areas [26]. In countries where significant proportions of the population live in rural areas, there was no statistically significant association between area of residence and adverse outcomes [46, 67, 68].

Fifteen studies [34, 37, 40, 42, 43, 45, 47–49, 52, 54, 64, 68, 86, 96] reported variations in neonatal mortality based on geographical/administrative location, but this was attributed to underlying factors such as high levels of poverty, underinvestment in health, and related infrastructure.

## Discussion

Our study sought to identify proximate and distant factors that influence maternal and neonatal mortality in LMICs. To our knowledge, this is the first study to comprehensively compile the evidence around predictors of newborn and maternal mortality in LMIC. Importantly, although we aimed to explore factors that influence both maternal and neonatal mortality, we identified very few studies exploring maternal mortality that met inclusion criteria.

The following proximate factors were associated with an increased risk of neonatal mortality: young or older maternal age, parity, short birth interval, first and higher birth order/rank, male neonate, extremities of birth weight, multiple-gestation, previous history of death of child, and lack of or inadequate ANC visits. This is consistent with findings from other studies [100–104]. The proximate factors associated with an increased risk of maternal mortality include maternal age, birth interval, parity, place of delivery, and inadequate ANC.

Distant factors associated with higher risks of neonatal mortality, which are consistent with other studies [104–107], included: low levels of maternal and paternal education, employment of mother and/or male partner in informal sectors such as agriculture, being resident in rural areas, low household income, use of solid fuels, and lack of clean water. Distant factors associated with higher risks of maternal mortality included maternal education, marital status, household income, resident of rural areas, and intimate partner violence. Marital status was reported to be significantly associated with maternal mortality in Asamoah's study [26], but had a weak association with neonatal mortality, which may indicate stronger influence of other proximate factors in determining postnatal outcomes.

We note that facility deliveries, skilled birth attendance, and desire for pregnancy did not have strong association with neonatal mortality in part due to presence of confounders. Facility deliveries appear to be associated with a lower risk of neonatal mortality. However, six studies reported lower risk for home births, which the authors attributed to low prevalence of facility deliveries in their study population. These findings might also be attributed to women with delivery complications being referred to health facilities. The observed lower risk of neonatal deaths with no skilled birth attendant versus physician-attended deliveries was not surprising given that in LMICs, physicians usually attend only to women with complications. A more appropriate comparison for further research efforts would compare midwife-supported deliveries with no skilled birth attendant.

These findings point to implications for programs and policy. Many of the identified risk factors can be identified during pregnancy and/or the postnatal period by facility- and community-based health workers, allowing for prioritization of mother-baby dyads for closer observation and/or home visits. Since most mortality occurs within the first week post-delivery, identifying relevant risk factors early in pregnancy could allow for a better plan of care throughout the continuum for those with risk factors. While ANC coverage is generally higher in LMICs [108], efforts to strengthen ANC quality, continuity of care, and systems for detection and referral of at-risk mother-baby dyads are needed to facilitate early identification and action. However, as other factors may be identifiable only after delivery (e.g. place and mode of delivery, complications), pre-discharge screening for mothers delivering in health facilities needs to be coupled with community identification, such as through CHWs. Other factors may be infeasible for identification by CHWs (e.g., body mass index, maternal nutritional status), or may result in stigma (e.g., income, employment status, tribe/caste). As Muriuki et al [109] note, risk factors will need to be selected with consideration of relevance in the context, and feasibility for identification. Importantly, although clinical factors were not identified as part of this scoping review, any risk screening for mother-baby dyads must also include known clinical risk factors (e.g., heavy bleeding in delivery). Although most risk factors

identified in this scoping review were associated with neonatal mortality, a targeted home visit to the newborn should always include care for both the newborn and the mother.

A targeted PNC approach would require strengthening of key health system functions in many settings. A structured and functional relationship between community- and facility-based health workers is required to ensure effective risk identification and intervention. Known system-level barriers to PNC use (high rates of home deliveries, limited staffing of facility- and community-based health providers, infrastructure) [9] must be addressed to ensure sufficient resourcing for timely identification of mother-baby dyads with risk factors, and referral for quality follow-up care.

Although the study identified several risk factors associated with increased neonatal mortality, it is important to note that all pregnancies and deliveries are inherently high-risk events; adverse outcomes may occur even in mother-baby dyads with no identifiable risk. A targeted PNC approach should be considered only in the context of broader efforts to improve coverage and quality of PNC, so that mother-baby dyads without identifiable risk factors receive timely PNC [109]. These findings also point to risk factors that should be mitigated; comprehensive, multisectoral "upstream" efforts to reduce the prevalence of risk factors will lead to improvements in maternal and newborn outcomes.

Findings from this scoping review also point to gaps for further research. Emerging factors such as gender dynamics at household level, role of access to information through media, climate change, humanitarian, and conflict settings, remain understudied and should be explored in future research efforts. We note variations in categorization and cut-off definitions for several factors, including age, parity, birth order, and education. Lack of standardized categorization and reference points for the various factors not only complicated comparison of findings across studies, but also influenced the direction of the observed relationship. Using maternal age as an example, various studies used different age bands to categorize this factor and then compared the outcomes with different reference age groups, thus obtaining varying results at multivariate analysis. Standardized categorizations would be useful in strengthening the body of evidence and allow for comparisons across various settings. In addition, more appropriate comparative studies to better understand the directionality of facility delivery in mitigating maternal and neonatal mortality are needed; as noted above, skilled birth attendance may be a more appropriate indicator in LMIC settings.

## Limitations

We note several limitations inherent in the methods of scoping review. Our categorization of studies did not consider study rigor or sample size. Excluding sub-national studies could mean excluding potentially relevant regional associations not found using national-level data. No efforts were made to assess associations by country and/or region, which may have missed important contextual variations in identified associations. It is possible that we missed some relevant studies despite our best efforts to conduct a comprehensive review. The cross-sectional nature of included studies limits the establishment of cause-effect relationships.

We noted several limitations in the studies identified. As noted, only five papers out of the seventy-nine met the inclusion criteria for maternal mortality because many of the studies that reported on maternal mortality were at sub-national and/or facility level and did not meet inclusion criteria. Furthermore, there was a dearth of studies that looked at the mother-baby dyad as a unit of study. There was also paucity of studies that assessed the risk factors at the various stages of the postnatal period, an important gap given the varying risk of mortality within the six-week postnatal period. The exclusion of facility-level papers inevitably led to exclusion of clinical factors, which are equally important and must be included in efforts to

screen and identify mother-baby dyads for targeted home visits. However, facility-based studies would have selected for mothers with access to a health facility; population-based studies also corrected for underlying factors influencing access. We note that the role of some important factors such as gender dynamics at the household level, the role of health insurance, maternal/household exposure to media, etc., were assessed in very few studies and provide an opportunity for more research in these areas. In communities where there was homogeneity in an assessed factor, its effect on maternal and neonatal mortality appeared to be less significant. For example, in studies where most participants resided in rural areas, the effect of residence on maternal and neonatal mortality was less significant than in those where there were significant proportions residing in urban and rural areas. Last, due to the lack of standardization of methodologies and/or approaches, the data were not amenable to meta-analysis, and we could not consider strength of association due to varied methodologies across studies.

## Conclusions

Our paper presents proximate and distant determinants of health associated with maternal and neonatal mortality in LMICs. Although many of these factors have been studied previously and most of our findings are consistent with other studies, they continue to play a significant role in adverse outcomes. These findings should help policy makers and program implementers identify and prioritize at-risk mother-baby dyads for targeted care to better their outcomes, while ensuring quality, timely PNC for all mothers and babies. Further, more efforts are needed to reduce the prevalence of these risk factors, including broader efforts to improve the broader socio-economic status at community level. While we aimed to explore factors associated with maternal mortality, we found very limited studies that met inclusion criteria. Investments in population-based studies to identify factors associated with maternal mortality are needed. Harmonization of categorization of factors such as age and education are another critical gap for consideration in future efforts.

## Supporting information

**S1 Checklist. Preferred Reporting Items for Systematic reviews and Meta-Analyses extension for Scoping Reviews (PRISMA-ScR) checklist.**
(DOCX)

**S1 Fig. Mosley and Chen framework.**
(TIF)

**S1 File. Scoping review search terms.**
(DOCX)

**S2 File. Protocol.**
(DOCX)

## Acknowledgments

The authors would like to acknowledge the Save the Children US team of Courtney James, Rudo Kashiri, and Amy Mangieri for their support during the review; Dr. Sarah Elaraby and Rachel Dean for their support for updating the review; and Erin Shea for her support in preparing this manuscript.

## Author Contributions

**Conceptualization:** Angela Muriuki, Melanie Yahner, Virginia Fonner, Syeda Nabin Ara Nitu, Joseph de Graft-Johnson.

**Data curation:** Preston Izulla, Michael Kiragu, Bernard Osir, Farahat Bello.

**Formal analysis:** Preston Izulla, Angela Muriuki, Michael Kiragu, Bernard Osir, Farahat Bello.

**Investigation:** Preston Izulla, Michael Kiragu.

**Methodology:** Preston Izulla, Angela Muriuki, Virginia Fonner.

**Project administration:** Preston Izulla, Bernard Osir.

**Supervision:** Preston Izulla, Michael Kiragu, Virginia Fonner.

**Visualization:** Bernard Osir.

**Writing – original draft:** Preston Izulla, Angela Muriuki, Michael Kiragu, Melanie Yahner, Virginia Fonner, Syeda Nabin Ara Nitu, Joseph de Graft-Johnson.

**Writing – review & editing:** Preston Izulla, Angela Muriuki, Michael Kiragu, Melanie Yahner, Farahat Bello, Joseph de Graft-Johnson.

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
