## [Decision Letter · Decision Letter 0]

27 Jul 2023

PONE-D-23-16950Proximate and distant determinants of maternal and neonatal mortality in the postnatal period: A scoping review of data from low- and middle-income countriesPLOS ONE

Dear Dr. Yahner,

Thank you for submitting your manuscript to PLOS ONE. After careful consideration, we feel that it has merit but does not fully meet PLOS ONE’s publication criteria as it currently stands. Therefore, we invite you to submit a revised version of the manuscript that addresses the points raised during the review process.

We look forward to receiving your revised manuscript.

Kind regards,

Seifadin Ahmed Shallo, MPH

Academic Editor

PLOS ONE

Journal Requirements:

 Financial support for the research, authorship and publication of this article was received through a grant from the Bill & Melinda Gates Foundation (INV-010128). The findings and conclusions contained within are those of the authors and do not necessarily reflect positions or policies of the Bill & Melinda Gates Foundation.

Reviewers' comments:

Reviewer's Responses to Questions

**Comments to the Author**

1. Is the manuscript technically sound, and do the data support the conclusions?

Reviewer #1: Yes

2. Has the statistical analysis been performed appropriately and rigorously? 

Reviewer #1: N/A

3. Have the authors made all data underlying the findings in their manuscript fully available?

Reviewer #1: Yes

4. Is the manuscript presented in an intelligible fashion and written in standard English?

Reviewer #1: Yes

5. Review Comments to the Author

Reviewer #1: Proximate and distant determinants of maternal and neonatal mortality in the postnatal period: A scoping review of data from low- and middle-income countries

Major comments:

• In the methodology section, kindly elaborate the methodological Quality assessment.

• There is a discrepancy in the time line mentioned in the methodology and supplementary material provided. Kindly clarify.

• I think, multiple gestation may come under Maternal factors, but not under neonatal factors. Authors are requested to relook into this particular variable and make amendments accordingly.

• Since authors have included multiple study designs, it is suggested to categorize and report number of studies included (E.g. No. of analytical cross-sectionals studies included etc).

• References are missing in the results section. For instance, where ever facts are produced from various studies reported under Proximate and distant factors.

Minor Comments:

• At line 90-92, there is a repetition of point. Please delete.

• In the methods section, please indicate whether a review protocol exists. If available, please provide registration details, including registration number.

• Present the full electronic search strategy for at least one database, including any limits used in the methods section.

6. PLOS authors have the option to publish the peer review history of their article (what does this mean?). If published, this will include your full peer review and any attached files.

Reviewer #1: No

---

## [Author Response · Author response to Decision Letter 0]

11 Oct 2023

Initial comments received in July 2023

Reviewer 1

Comment: References are missing in the results section. For examples, where facts are produced from various studies reported under Proximate and distant factors. 

Response: We have added references throughout the Results section. 

Comment: At line 90-92, there is a repetition of point. Please delete. 

Response: Thank you. We have removed the repeated sentence.

Comment: In the methods section, please indicate whether a review protocol exists. If available, please provide registration details, including registration number. 

Response: We initially developed a protocol for a systematic review, and registered it on PROSPERO with registration number CRD42021234905. However, we shifted to a scoping review because our research question were broad; we were looking at many varying factors affecting the outcome(s) of interest. Our search criteria were flexible, and evolved based on our initial findings in line with the iterative nature of a scoping review.

We have included the original protocol as supporting information, and we are available to provide further information as needed.

Comment: Present the full electronic search strategy for at least one database, including any limits used in the methods section. 

Response: We have added detail in the Methods section (Line 105) and clarified that limits are listed in Table 1. The added detail on the methodological quality assessment (see point below) also provides relevant detail. Supplement 1 does provide the full search strategy. Please do advise if further information is necessary.

Comment: Methodological Quality assessment is not clearly mentioned. Authors are requested to elaborate in this part of methods. 

Response: We appreciate this suggestion. We have added detail in the Methods section (beginning at Line 134).

Comment: Time line mentioned in the methodology and supplementary material are different. Kindly clarify. 

Response: Thank you for catching this discrepancy. We have updated the supplementary material to reflect the correct time line (2011-2022).

Comment: Multiple gestation may come under Maternal factors but not under neonatal factors. Authors are requested to relook into this particular variable. 

Response: Thank you. We have moved the information about Multiple Gestation to the section on neonatal factors.

Comment: Since authors have included multiple study designs, it is suggested to categorize number of studies included and reported (E.g., no of analytical cross-sectionals studies included). 

Response: We appreciate this suggestion. We have added information about the categorization of the included studies beginning on Line 182.

Editorial team comments

Comment: Please ensure that your manuscript meets PLOS ONE's style requirements, including those for file naming. The PLOS ONE style templates can be found here and here 

Response: Thank you. We have reformatted throughout, and standardized the file names.

Comment: Thank you for stating the following financial disclosure: 

Financial support for the research, authorship and publication of this article was received through a grant from the Bill & Melinda Gates Foundation (INV-010128). The findings and conclusions contained within are those of the authors and do not necessarily reflect positions or policies of the Bill & Melinda Gates Foundation.

Response: Thank you. We have included the suggested statement in our cover letter. We appreciate your offer to update the online submission form on our behalf.

Comment: In your Data Availability statement, you have not specified where the minimal data set underlying the results described in your manuscript can be found. PLOS defines a study's minimal data set as the underlying data used to reach the conclusions drawn in the manuscript and any additional data required to replicate the reported study findings in their entirety. All PLOS journals require that the minimal data set be made fully available. For more information about our data policy, please see here.

"Upon re-submitting your revised manuscript, please upload your study’s minimal underlying data set as either Supporting Information files or to a stable, public repository and include the relevant URLs, DOIs, or accession numbers within your revised cover letter. For a list of acceptable repositories, please see here. Any potentially identifying patient information must be fully anonymized.

Important: If there are ethical or legal restrictions to sharing your data publicly, please explain these restrictions in detail. Please see our guidelines for more information on what we consider unacceptable restrictions to publicly sharing data:. Note that it is not acceptable for the authors to be the sole named individuals responsible for ensuring data access.

Response: As this is a scoping review, we do not have primary data to make available. All data included are technically already publicly available, as the data were extracted from published literature. Original sources are available through the references, and this paper represents the full findings of our scoping review. We have updated this information in the Data Availability statement, and welcome alternative guidance.

Comment: Please include captions for your Supporting Information files at the end of your manuscript, and update any in-text citations to match accordingly. Please see our Supporting Information guidelines for more information:. 

Response: We have updated and standardized the Supporting Information files themselves, and updated the in-text citations.

Comment: Please review your reference list to ensure that it is complete and correct. If you have cited papers that have been retracted, please include the rationale for doing so in the manuscript text, or remove these references and replace them with relevant current references. Any changes to the reference list should be mentioned in the rebuttal letter that accompanies your revised manuscript. If you need to cite a retracted article, indicate the article’s retracted status in the References list and also include a citation and full reference for the retraction notice.

Response: To the best of our knowledge, our reference list is complete and correct, and does not include papers that have been retracted.

Comments received on October 3

Comment 1. Please upload a Response to Reviewers letter which should include a point by point response to each of the points made by the Editor and / or Reviewers. (This should be uploaded as a 'Response to Reviewers' file type.) Please follow this link for more information: https://nam12.safelinks.protection.outlook.com/?url=http%3A%2F%2Fblogs.plos.org%2Feveryone%2F2011%2F05%2F10%2Fhow-to-submit-your-revised-manuscript%2F&data=05%7C01%7Cmyahner%40savechildren.org%7Cce062338d0c3424694fc08dbc448b2a9%7Cd1934b2d792c47cca2f5fc634183cd2d%7C0%7C0%7C638319588781151501%7CUnknown%7CTWFpbGZsb3d8eyJWIjoiMC4wLjAwMDAiLCJQIjoiV2luMzIiLCJBTiI6Ik1haWwiLCJXVCI6Mn0%3D%7C3000%7C%7C%7C&sdata=QDbq65%2B5zFbIydkNQh9IwZHSALO26X3QuEeDWEUwbBU%3D&reserved=0

Response: Thank you. We have uploaded an updated Response to Reviewers letter, with a point-by-point response to each comment.

Comment 2. Please note that funding information should not appear in the Acknowledgments section or other areas of your manuscript. We will only publish funding information present in the Funding Statement section of the online submission form. Please remove any funding-related text from the manuscript.

Response: Thank you. We have removed the funding information that appeared in the body of the manuscript.

Comment 3. Can you please upload an additional copy of your revised manuscript that does not contain any tracked changes or highlighting as your main article file. This will be used in the production process if your manuscript is accepted. Please amend the file type for the file showing your changes to Revised Manuscript w/tracked changes. Please follow this link for more information: https://nam12.safelinks.protection.outlook.com/?url=http%3A%2F%2Fblogs.plos.org%2Feveryone%2F2011%2F05%2F10%2Fhow-to-submit-your-revised-manuscript%2F&data=05%7C01%7Cmyahner%40savechildren.org%7Cce062338d0c3424694fc08dbc448b2a9%7Cd1934b2d792c47cca2f5fc634183cd2d%7C0%7C0%7C638319588781151501%7CUnknown%7CTWFpbGZsb3d8eyJWIjoiMC4wLjAwMDAiLCJQIjoiV2luMzIiLCJBTiI6Ik1haWwiLCJXVCI6Mn0%3D%7C3000%7C%7C%7C&sdata=QDbq65%2B5zFbIydkNQh9IwZHSALO26X3QuEeDWEUwbBU%3D&reserved=0

Response: Thank you. We have uploaded a clean version of the revised manuscript.

Comment 4. Please ensure that you refer to Table 1 in your text as, if accepted, production will need this reference to link the reader to the Table.

Response: Thank you. We have added a line to introduce Table 1 in the text (Line 116).

---

## [Editor Report · Decision Letter 1]

13 Oct 2023

Proximate and distant determinants of maternal and neonatal mortality in the postnatal period: A scoping review of data from low- and middle-income countries

PONE-D-23-16950R1

Dear Dr. Yahner,

We’re pleased to inform you that your manuscript has been judged scientifically suitable for publication and will be formally accepted for publication once it meets all outstanding technical requirements.

Kind regards,

Seifadin Ahmed Shallo, MPH

Academic Editor

PLOS ONE

---

## [Editor Report · Acceptance letter]

27 Oct 2023

PONE-D-23-16950R1 

Proximate and distant determinants of maternal and neonatal mortality in the postnatal period: A scoping review of data from low- and middle-income countries 

Dear Dr. Yahner:

I'm pleased to inform you that your manuscript has been deemed suitable for publication in PLOS ONE. Congratulations! Your manuscript is now with our production department. 

Kind regards, 

on behalf of

Prof. Seifadin Ahmed Shallo 

Academic Editor

PLOS ONE